# Controlling privacy in recommender systems

**Yu Xin**
CSAIL, MIT
yuxin@mit.edu

**Tommi Jaakkola**
CSAIL, MIT
tommi@csail.mit.edu

## Abstract

Recommender systems involve an inherent trade-off between accuracy of recommendations and the extent to which users are willing to release information about their preferences. In this paper, we explore a two-tiered notion of privacy where there is a small set of "public" users who are willing to share their preferences openly, and a large set of "private" users who require privacy guarantees. We show theoretically and demonstrate empirically that a moderate number of public users with no access to private user information already suffices for reasonable accuracy. Moreover, we introduce a new privacy concept for gleaning relational information from private users while maintaining a first order deniability. We demonstrate gains from controlled access to private user preferences.

## 1 Introduction

Recommender systems exploit fragmented information available from each user. In a realistic system there's also considerable "churn", i.e., users/items entering or leaving the system. The core problem of transferring the collective experience of many users to an individual user can be understood in terms of matrix completion ([13, 14]). Given a sparsely populated matrix of preferences, where rows and columns of the matrix correspond to users and items, respectively, the goal is to predict values for the missing entries.

Matrix completion problems can be solved as convex regularization problems, using trace norm as a convex surrogate to rank. A number of algorithms are available for solving large-scale trace-norm regularization problems. Such algorithms typically operate by iteratively building the matrix from rank-1 components (e.g., [7, 17]). Under reasonable assumptions (e.g., boundedness, noise, restricted strong convexity), the resulting empirical estimators have been shown to converge to the underlying matrix with high probability ([12, 8, 2]). Consistency guarantees have mostly involved matrices of fixed dimension, i.e., generalization to new users is not considered. In this paper, we reformulate the regularization problem in a manner that depends only on the item (as opposed to user) features, and characterize the error for out-of-sample users.

The completion accuracy depends directly on the amount of information that each user is willing to share with the system ([1]). It may be possible in some cases to side-step this statistical trade-off by building Peer-to-Peer networks with homomorphic encryption that is computationally challenging([3, 11]). We aim to address the statistical question directly.

The statistical trade-off between accuracy and privacy further depends on the notion of privacy we adopt. A commonly used privacy concept is Differential Privacy (DP) ([6]), first introduced to protect information leaked from database queries. In a recommender context, users may agree to a trusted party to hold and aggregate their data, and perform computations on their behalf. Privacy guarantees are then sought for any results published beyond the trusted party (including back to the users). In this setting, differential privacy can be achieved through obfuscation (adding noise) without a significant loss of accuracy ([10]).

In contrast to [10], we view the system as an untrusted entity, and assume that users wish to guard their own data. We depart from differential privacy and separate computations that can be done locally (privately) by individual users and computations that must be performed by the system (e.g., aggregation). For example, in terms of low rank matrices, only the item features have to be solved by the system. The corresponding user features can be obtained locally by the users and subsequently used for ranking.

From this perspective, we divide the set of users into two pools, the set of public users who openly share their preferences, and the larger set of private users who require explicit privacy guarantees. We show theoretically and demonstrate empirically that a moderate number of public users suffice for accurate estimation of item features. The remaining private users can make use of these item features without any release of information. Moreover, we propose a new 2nd order privacy concept which uses limited (2nd order) information from the private users as well, and illustrate how recommendations can be further improved while maintaining marginal deniability of private information.

## 2   Problem formulation and summary of results

**Recommender setup without privacy**   Consider a recommendation problem with $n$ users and $m$ items. The underlying complete rating matrix to be recovered is $\mathring{X} \in \mathbb{R}^{n \times m}$. If only a few latent factors affect user preferences, $\mathring{X}$ can be assumed to have low rank. As such, it is also recoverable from a small number of observed entries. We assume that entries are observed with noise. Specifically,

$$Y_{i,j} = \mathring{X}_{i,j} + \epsilon_{i,j}, \ \ (i,j) \in \Omega \tag{1}$$

where $\Omega$ denotes the set of observed entries. Noise is assumed to be $i.i.d$ and follows a zero-mean sub-Gaussian distribution with parameter $\|\epsilon\|_{\psi_2} = \sigma$. Following [16], we refer to the noise distribution as $Sub(\sigma^2)$.

To bias our estimated rating matrix $X$ to have low rank, we use convex relaxation of rank in the form of trace norm. The trace-norm is the sum of singular values of the matrix or $\|X\|_* = \sum_i \sigma_i(X)$. The basic estimation problem, without any privacy considerations, is then given by

$$\min_{X \in \mathbb{R}^{m \times n}} \ \frac{1}{N} \sum_{(i,j) \in \Omega} (Y_{i,j} - X_{i,j})^2 + \frac{\lambda}{\sqrt{mn}} \|X\|_* \tag{2}$$

where $\lambda$ is a regularization parameter and $N = |\Omega|$ is the total number of observed ratings. The factor $\sqrt{mn}$ ensures that the regularization does not grow with either dimension.

The above formulation requires the server to explicitly obtain predictions for each user, i.e., solve for $X$. We can instead write $X = UV^T$ and $\Sigma = (1/\sqrt{mn})VV^T$, and solve for $\Sigma$ only. If the server then communicates the resulting low rank $\Sigma$ (or just $V$) to each user, the users can reconstruct the relevant part of $U$ locally, and reproduce $X$ as it pertains to them. To this end, let $\phi_i = \{j : (i,j) \in \Omega\}$ be the set of observed entries for user $i$, and let $Y_{i,\phi_i}$ be a column vector of user $i$'s ratings. Then we can show that Eq.(2) is equivalent to solving

$$\min_{\Sigma \in S^+} \ \sum_{i=1}^{n} Y_{i,\phi_i}^T (\lambda' I + \Sigma_{\phi_i,\phi_i}) Y_{i,\phi_i} + \sqrt{nm} \, \|\Sigma\|_* \tag{3}$$

where $S^+$ is the set of positive semi-definite $m \times m$ matrices and $\lambda' = \lambda N/\sqrt{nm}$. By solving $\hat{\Sigma}$, we can predict ratings for unobserved items (index set $\phi_i^c$ for user $i$) by

$$\hat{X}_{i,\phi_i^c} = \Sigma_{\phi_i^c,\phi_i} (\lambda' I + \Sigma_{\phi_i,\phi_i})^{-1} Y_{i,\phi_i} \tag{4}$$

Note that we have yet to address any privacy concerns. The solution to Eq.(3) still requires access to full ratings $Y_{i,\phi_i}$ for each user.

**Recommender setup with privacy**   Our privacy setup assumes an untrusted server. Any user interested in guarding their data must therefore keep and process their data locally, releasing information to the server only in a controlled manner. We will initially divide users into two broad

categories, public and private. Public users are willing to share all their data with the server while private users are unwilling to share any. This strict division is removed later when we permit private users to release, in a controlled manner, limited information pertaining to their ratings (2nd order information) so as to improve recommendations.

Any data made available to the server enables the server to model the collective experience of users, for example, to solve Eq.(3). We will initially consider the setting where Eq.(3) is solved on the basis of public users only. We use an EM type algorithm for training. In the E-step, the current $\Sigma$ is sent to public users to complete their rating vectors and send back to the server. In the M-step, $\Sigma$ is then updated based on these full rating vectors. The resulting $\hat{\Sigma}$ (or $\hat{V}$) can be subsequently shared with the private users, enabling the private users (their devices) to locally rank candidate items without any release of private information. The estimation of $\hat{\Sigma}$ is then improved by asking private users to share 2nd order relational information about their ratings without any release of marginal selections/ratings.

Note that we do not consider privacy beyond ratings. In other words, we omit any subsequent release of information due to users exploring items recommended to them.

**Summary of contributions** We outline here our major contributions towards characterizing the role of public users and the additional controlled release of information from private users.

**1)** We show that $\mathring{\Sigma} = \sqrt{\mathring{X}^T \mathring{X}}/\sqrt{nm}$ can be estimated in a consistent, accurate manner on the basis of public users alone. In particular, we express the error $\|\hat{\Sigma} - \mathring{\Sigma}\|_F$ as a function of the total number of observations. Moreover, if the underlying public user ratings can be thought of as i.i.d. samples, we also bound $\|\mathring{\Sigma} - \Sigma^*\|_F$ in terms of the number of public users. Here $\Sigma^*$ is the true limiting estimate. See section 3.1 for details.

**2)** We show how the accuracy of predicted ratings $\hat{X}_{i,\phi_i^c}$ for private users relates to the accuracy of estimating $\hat{\Sigma}$ (primarily from public users). Since the ratings for user $i$ may not be related to the subspace that $\hat{\Sigma}$ lies in, we can only characterize the accuracy when sufficient overlap exists. We quantify this overlap, and show how $\|\hat{X}_{i,\phi_i^c} - \mathring{X}_{i,\phi_i^c}\|$ depends on this overlap, accuracy of $\hat{\Sigma}$, and the observation noise. See section 3.2 for details.

**3)** Having established the accuracy of predictions based on public users alone, we go one step further and introduce a new privacy mechanism and algorithms for gleaning additional relational (2nd order) information from private users. This 2nd order information is readily used by the server to estimate $\hat{\Sigma}$. The privacy concept constructively maintains first order (marginal) deniability for private users. We demonstrate empirically the gains from the additional 2nd order information. See section 4.

# 3 Analysis

## 3.1 Statistical Consistency of $\hat{\Sigma}$

Let $\hat{X}$ be a solution to Eq.(2). We can write $\hat{X} = \hat{U}\hat{V}^T$, where $\hat{U}^T\hat{U} = \hat{I}_m$ with 0/1 diagonal. Since $\hat{\Sigma} = \frac{1}{\sqrt{mn}}\sqrt{\hat{X}^T\hat{X}}$ we can first analyze errors in $\hat{X}$ and then relate them to $\hat{\Sigma}$. To this end, we follow the restricted strong convexity (RSC) analysis[12]. However, their result depends on the inverse of the minimum number of ratings of all users and items. In practice (see below), the number of ratings decays exponentially across sorted users, making such a result loose. We provide a modified analysis that depends only on the total number of observations $N$.

Throughout the analysis, we assume that each row vector $\mathring{X}_{i,.}$ belongs to a fixed $r$ dimensional subspace. We also assume that both noiseless and noisy entries are bounded, i.e. $|Y_{i,j}|, |\mathring{X}_{i,j}| \leq \alpha, \forall (i,j)$. For brevity, we use $\|Y - X\|_\Omega^2$ to denote the empirical loss $\sum_{(i,j)\in\Omega}(Y_{i,j} - X_{i,j})^2$. The restricted strong convexity property (RSC) assumes that there exists a constant $\kappa > 0$ such that

$$\frac{\kappa}{mn}\|\hat{X} - \mathring{X}\|_F^2 \leq \frac{1}{N}\|\hat{X} - \mathring{X}\|_\Omega^2 \tag{5}$$

for $\hat{X} - \mathring{X}$ in a certain subset. RSC provides the step from approximating observations to approximating the full underlying matrix. It is satisfied with high probability provided that $N = (m + n)\log(m + n))$.

Assume the SVD of $\mathring{X} = \mathring{P}S\mathring{Q}^T$, and let row(X) and col($X$) denote the row and column spaces of $X$. We define the following two sets,

$$
\begin{aligned}
A(P, Q) &:= \{X, \text{row}(X) \subseteq \mathring{P}, \text{col}(X) \subseteq \mathring{Q}\} \\
B(P, Q) &:= \{X, \text{row}(X) \subseteq \mathring{P}^\perp, \text{col}(X) \subseteq \mathring{Q}^\perp\}
\end{aligned}
\tag{6}
$$

Let $\pi_A(X)$ and $\pi_B(X)$ be the projection of $X$ onto sets $A$ and $B$, respectively, and $\pi_{\overline{A}} = I - \pi_A$, $\pi_{\overline{B}} = I - \pi_B$. Let $\Delta = \hat{X} - \mathring{X}$ be the difference between the estimated and the underlying rating matrices. Our first lemma demonstrates that $\Delta$ lies primarily in a restricted subspace and the second one guarantees that the noise remains bounded.

**Lemma 3.1.** *Assume $\epsilon_{i,j}$ for $(i, j) \in \Omega$ are i.i.d. sub-gaussian with $\sigma = \|\epsilon_{i,j}\|_{\psi_1}$. Then with probability $1 - \frac{e}{N^{4ch}}$, $\|\pi_B(\Delta)\|_* \leq \|\pi_{\overline{B}}(\Delta)\|_* + \frac{2c^2\sigma^2\sqrt{mn}}{N\lambda}\log^2 N$. Here $h > 0$ is an absolute constant associated with the sub-gaussian noise.*

If $\lambda = \lambda_0 c\sigma \frac{\log^2 N}{\sqrt{N}}$, then $\frac{c^2\sigma^2\sqrt{mn}\log N}{N\lambda} = \frac{c\sigma \log N}{\lambda_0}\sqrt{\frac{mn}{N}} = b\log N\sqrt{\frac{n}{N}}$ where we leave the dependence on $n$ explicit. Let $\mathcal{D}(b, n, N)$ denote the set of difference matrices that satisfy lemma 3.1 above. By combining the lemma and the RSC property, we obtain the following theorem.

**Theorem 3.2.** *Assume RSC for the set $D(b, n, N)$ with parameter $\kappa > 0$ where $b = \frac{c\sigma\sqrt{m}}{\lambda_0}$. Let $\lambda = \lambda_0 c\sigma \frac{\log N}{\sqrt{N}}$, then we have $\frac{1}{\sqrt{mn}}\|\Delta\|_F \leq 2c\sigma(\frac{1}{\sqrt{\kappa}} + \frac{\sqrt{2r}}{\kappa})\frac{\log N}{\sqrt{N}}$ with probability at least $1 - \frac{e}{N^{4ch}}$ where $h, c > 0$ are constants.*

The bound in the theorem consists of two terms, pertaining to the noise and the regularization. In contrast to [12], the terms only relate to the total number of observations $N$.

We now turn our focus on the accuracy of $\hat{\Sigma}$. First, we map the accuracy of $\hat{X}$ to that of $\hat{\Sigma}$ using a perturbation bound for polar decomposition (see [9]).

**Lemma 3.3.** *If $\frac{1}{\sqrt{mn}}\|\hat{X} - \mathring{X}\|_F \leq \delta$, then $\|\hat{\Sigma} - \mathring{\Sigma}\|_F \leq \sqrt{2}\delta$*

This completes our analysis in terms of recovering $\mathring{\Sigma}$ for a fixed size underlying matrix $\mathring{X}$. As a final step, we turn to the question of how the estimation error changes as the number of users or $n$ grows. Let $\mathring{X}_i$ be the underlying rating vector for user $i$ and define $\Theta^n = \frac{1}{mn}\sum_{i=1}^n \mathring{X}_i^T \mathring{X}_i$. Then $\mathring{\Sigma} = (\Theta^n)^{\frac{1}{2}}$. If $\Theta^*$ is the limit of $\Theta^n$, then $\Sigma^* = (\Theta^*)^{\frac{1}{2}}$. We bound the distance between $\mathring{\Sigma}$ and $\Sigma^*$.

**Theorem 3.4.** *Assume $\mathring{X}_i$ are i.i.d samples from a distribution with support only in a subspace of dimension $r$ and bounded norm $\|\mathring{X}_i\| \leq \alpha\sqrt{m}$. Let $\beta_1$ and $\beta_r$ be the smallest and largest eigenvalues of $\Sigma^*$. Then, for large enough $n$, with probability at least $1 - \frac{r}{n^2}$,*

$$
\|\mathring{\Sigma} - \Sigma^*\|_F \leq 2\sqrt{r}\alpha\sqrt{\frac{\beta_r \log n}{\beta_1 n}} + o(\frac{\log n}{n})
\tag{7}
$$

Combining the two theorems and using triangle inequality, we obtain a high probability bound on $\|\hat{\Sigma} - \Sigma^*\|_F$. Assuming the number of ratings for each user is larger than $\xi m$, then $N > \xi nm$ and the bound grows in the rate of $\eta(\log n / \sqrt{n})$ with $\eta$ being a constant that depends on $\xi$. For large enough $\xi$, the required $n$ to achieve a certain error bound is small. Therefore a few public users with large number of ratings could be enough to obtain a good estimate of $\Sigma^*$.

## 3.2 Prediction accuracy

We are finally ready to characterize the error in the predicted ratings $\hat{X}_{i,\phi_i^c}$ for all users as defined in Eq.(4). For brevity, we use $\delta$ to represent the bound on $\|\hat{\Sigma} - \Sigma^*\|$ obtained on the basis of our results above. We also use $x_\phi$ and $x_{\phi^c}$ as shorthands for $X_{i,\phi_i}$ and $X_{i,\phi_i^c}$ with the idea that $x_\phi$ typically refers to a new private user.

The key issue for us here is that the partial rating vector $x_\phi$ may be of limited use. For example, if the number of observed ratings is less than rank $r$, then we would be unable to identify a rating vector in the $r$ dimensional subspace even without noise. We seek to control this in our analysis by assuming that the observations have enough signal to be useful. Let SVD of $\Sigma^*$ be $Q^* S^* (Q^*)^T$, and $\beta_1$ be its minimum eigenvalue. We constrain the index set of observations $\phi$ such that it belongs to the set

$$\mathcal{D}(\gamma) = \left\{ \phi \subseteq \{1, \ldots, m\}, s.t. \|x\|_F^2 \leq \gamma \frac{m}{|\phi|} \|x_\phi\|_F^2, \forall x \in \text{row}((Q^*)^T) \right\}$$

The parameter $\gamma$ depends on how the low dimensional sub-space is aligned with the coordinate axes. We are only interested in characterizing prediction errors for observations that are in $\mathcal{D}(\gamma)$. This is quite different from the RSC property. Our main result is then

**Theorem 3.5.** *Suppose* $\|\Sigma - \Sigma^*\|_F \leq \delta \ll \beta_1$, $\phi \in \mathcal{D}(\gamma)$. *For any* $\mathring{x} \in \text{row}((Q^*)^T)$, *our observation* $x_\phi = \mathring{x}_\phi + \epsilon_\phi$ *where* $\epsilon_\phi \sim Sub(\sigma^2)$ *is the noise vector. The predicted ratings over the remaining entries are given by* $\hat{x}_{\phi^c} = \Sigma_{\phi^c, \phi} (\lambda' I + \Sigma_{\phi, \phi})^{-1} x_\phi$. *Then, with probability at least* $1 - \exp(-c_2 \min(c_1^4, \sqrt{|\phi|} c_1^2))$,

$$\|x_{\phi^c} - \mathring{x}_{\phi^c}\|_F \leq 2\sqrt{\lambda' + \delta} (\sqrt{\gamma \frac{m}{|\phi|}} + 1)(\frac{\|\mathring{x}\|_F}{\sqrt{\beta_1}} + \frac{2c_1 \sigma |\phi|^{\frac{1}{4}}}{\sqrt{\lambda'}})$$

*where* $c_1, c_2 > 0$ *are constants.*

All the proofs are provided in the supplementary material. The term proportional to $\|\mathring{x}\|_F / \sqrt{\beta_1}$ is due to the estimation error of $\Sigma^*$, while the term proportional to $2c_1 \sigma |\phi|^{\frac{1}{4}} / \sqrt{\lambda'}$ comes from the noise in the observed ratings.

# 4   Controlled privacy for private users

Our theoretical results already demonstrate that a relatively small number of public users with many ratings suffices for a reasonable performance guarantee for both public and private users. Empirical results (next section) support this claim. However, since public users enjoy no privacy guarantees, we would like to limit the required number of such users by requesting private users to contribute in a limited manner while maintaining specific notions of privacy.

**Definition 4.1. : Privacy preserving mechanism.** *Let* $\mathcal{M} : \mathbb{R}^{m \times 1} \to \mathbb{R}^{m \times 1}$ *be a random mechanism that takes a rating vector* $r$ *as input and outputs* $\mathcal{M}(r)$ *of the same dimension with* $j^{th}$ *element* $\mathcal{M}(r)_j$. *We say that* $\mathcal{M}(r)$ *is element-wise privacy preserving if* $Pr(\mathcal{M}(r)_j = z) = p(z)$ *for* $j = 1, ..., m$, *and some fixed distribution* $p$.

For example, a privacy preserving mechanism $\mathcal{M}(r)$ is element-wise private if its coordinates follow the same marginal distribution such as uniform. Note that such a mechanism can still release information about how different ratings interact (co-vary) which is necessary for estimation.

**Discrete values.** Assume that each element in $r$ and $\mathcal{M}(r)$ belongs to a discrete set $S$ with $|S| = K$. A natural privacy constraint is to insist that the marginal distribution of $\mathcal{M}(r)_j$ is uniform, i.e., $Pr(\mathcal{M}(r)_j = z) = 1/K$, for $z \in S$. A trivial mechanism that satisfies the privacy constraint is to select each value uniformly at random from $S$. In this case, the returned rating vector contributes nothing to the server model. Our goal is to design a mechanism that preserves useful 2nd order information.

We assume that a small number of public user profiles are available, from which we can learn an initial model parameterized by $(\mu, V)$, where $\mu$ is the item mean vector and $V$ is a low rank component of $\Sigma$. The server provides each private user the pair $(\mu, V)$ and asks, once, for a response $\mathcal{M}(r)$. Here $r$ is the user's full rating vector, completed (privately) with the help of the server model $(\mu, V)$.

The mechanism $\mathcal{M}(r)$ is assumed to be element-wise privacy preserving, thus releasing nothing about a single element in isolation. What information should it carry? If each user $i$ provided their full rating vector $r^i$, the server could estimate $\Sigma$ according to $\frac{1}{\sqrt{nm}} (\sum_{i=1}^n (r^i - \mu)(r^i - \mu)^T)^{\frac{1}{2}}$. Thus,

if $\mathcal{M}(r)$ preserves the second order statistics to the extent possible, the server could still obtain an accurate estimate of $\Sigma$.

Consider a particular user and their completed rating vector $r$. Let $\mathrm{P}(x) = \mathrm{Pr}(\mathcal{M}(r) = x)$. We select distribution $\mathrm{P}(x)$ by solving the following optimization problem geared towards preserving interactions between the ratings under the uniform marginal constraint.

$$\min_{\mathrm{P}} \quad \mathrm{E}_{x \sim \mathrm{P}} \|(x - \mu)(x - \mu)^T - (r - \mu)(r - \mu)^T\|_F^2$$
$$s.t. \quad \mathrm{P}(x_i = s) = 1/K, \; \forall i, \; \forall s \in S. \tag{8}$$

where $K = |S|$. The exact solution is difficult to obtain because the number of distinct assignments of $x$ is $K^m$. Instead, we consider an approximate solution. Let $x^1, ..., x^K \in \mathbb{R}^{m \times 1}$ be $K$ different vectors such that, for each $i$, $\{x_i^1, x_i^2, ..., x_i^K\}$ forms a permutation of $S$. If we choose $x$ with $\mathrm{Pr}(x = x^j) = 1/K$, then the marginal distribution of each element is uniform therefore maintaining element-wise privacy. It remains to optimize the set $x^1, ..., x^K$ to capture interactions between ratings.

We use a greedy coordinate descent algorithm to optimize $x^1, ..., x^K$. For each coordinate $i$, we randomly select a pair $x^p$ and $x^q$, and switch $x_i^p$ and $x_i^q$ if the objective function in (8) is reduced. The process is repeated a few times before we move on to the next coordinate. The algorithm can be implemented efficiently because each operation deals only with a single coordinate.

Finally, according to the mechanism, the private user selects one of $x^j$, $j = 1, \ldots, K$, uniformly at random and sends the discrete vector back to the server. Since the resulting rating vectors from private users are noisy, the server decreases their weight relative to the information from public users as part of the overall M-step for estimating $\Sigma$.

**Continuous values.** Assuming the rating values are continuous and unbounded, we require instead that the returned rating vectors follow the marginal distributions with a given mean and variance. Specifically, $\mathrm{E}[\mathcal{M}(r)_i] = 0$ and $\mathrm{Var}[\mathcal{M}(r)_i] = v$ where $v$ is a constant that remains to be determined. Note that, again, any specific element of the returned vector will not, in isolation, carry any information specific to the element.

As before, we search for the distribution $P$ so as to minimize the $L_2$ error of the second order statistics under marginal constraints. For simplicity, denote $r' = r - \mu$ where $r$ is the true completed rating vector, and $u_i = \mathcal{M}(r)_i$. The objective is given by

$$\min_{\mathrm{P}, v} \quad \mathrm{E}_{u \sim \mathrm{P}} \|uu^T - r'r'^T\|_F^2$$
$$s.t. \quad \mathrm{E}[u_i] = 0, \; \mathrm{Var}[u_i] = v, \; \forall i. \tag{9}$$

Note that the formulation does not directly constrain that $P$ has identical marginals, only that the means and variances agree. However, the optimal solution does, as shown next.

**Theorem 4.2.** *Let $z_i = sign(r_i')$ and $h = (\sum_{i=1}^m |r_i'|)/m$. The minimizing distribution of (9) is given by $Pr(u = zh) = Pr(u = -zh) = 1/2$.*

We leave the proof in the supplementary material. A few remarks are in order. The mechanism in this case is a two component mixture distribution, placing a probability mass on $sign(r')h$ (vectorized) and $-sign(r')h$ with equal probability. As a result, the server, knowing the algorithm that private users follow, can reconstruct $sign(r')$ up to an overall randomly chosen sign. Note also that the value of $h$ is computed from user's private rating vector and therefore releases some additional information about $r' = r - \mu$ albeit weakly. To remove this information altogether, we could use the same $h$ for all users and estimate it based on public users.

The privacy constraints will clearly have a negative impact on the prediction accuracy in comparison to having direct access to all the ratings. However, the goal is to improve accuracy beyond the public users alone by obtaining limited information from private users. While improvements are possible, the limited information surfaces in several ways. First, since private users provide no first order information, the estimation of mean rating values cannot be improved beyond public users. Second, the sampling method we use to enforce element-wise privacy adds noise to the aggregate second order information from which $V$ is constructed. Finally, the server can run the M-step with respect to the private users' information only once, whereas the original EM algorithm could entertain different completions for user ratings iteratively. Nevertheless, as illustrated in the next section, the algorithm can still achieve a good accuracy, improving with each additional private user.

## 5 Experiments

We perform experiments on the Movielens 10M dataset which contains 10 million ratings from 69878 users on 10677 movies. The test set contains 10 ratings for each user. We begin by demonstrating that indeed a few public users suffice for making accurate recommendations. Figure 1 left shows the test performance of both weighted (see [12]) and unweighted (uniform) trace norm regularization as we add users. Here users with most ratings are added first.

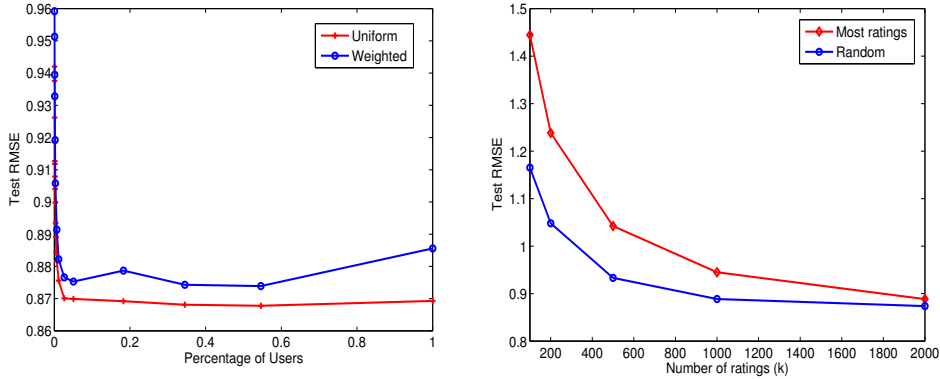

Figure 1: Left: Test RMSE as a function of the percentage of public users; Right: Performance changes with different rating numbers.

With only $1\%$ of public users added, the test RMSE of unweighted trace norm regularization is $0.876$ which is already close to the optimal prediction error. Note that the loss of weighted trace norm regularization actually starts to go up when the number of users increases. The reason is that the weighted trace norm penalizes less for users with few ratings. As a result, the resulting low dimensional subspace used for prediction is influenced more by users with few ratings.

The statistical convergence bound in section 3.1 involves both terms that decrease as a function of the number of ratings $N$ and the number of public users $n$. The two factors are usually coupled. It is interesting to see how they impact performance individually. Given a number of total ratings, we compare two different methods of selecting public users. In the first method, users with most ratings are selected first, whereas the second method selects users uniformly at random. As a result, if we equalize the total number of ratings from each method, the second method selects a lot more users. Figure 1 Right shows that the second method achieves better performance. An interpretation, based on the theory, is that the right side of error bound (7) decreases as the number of users increases.

We also show how performance improves based on controlled access to private user preferences. First, we take the top $100$ users with the most ratings as the public users, and learn the initial prediction model from their ratings. To highlight possible performance gains, private users with more ratings are selected first. The results remain close if we select private users uniformly.

The rating values are from $0.5$ to $5$ with totally $10$ different discrete values. Following the privacy mechanism for discrete values, each private user generates ten different candidate vectors and returns one of them uniformly at random. In the M-step, the weight for each private user is set to $1/2$ compared to 1 for public users. During training, after processing $w = 20$ private users, we update parameters $(\mu, V)$, re-complete the rating vectors of public users, making predictions for next batch of private users more accurate. The privacy mechanism for continuous values is also tested under the same setup. We denote the two privacy mechanism as PMD and PMC, respectively.

We compare five different scenarios. First, we use a standard DP method that adds Laplace noise to the completed rating vector. Let the DP parameter be $\epsilon$, because the maximum difference between rating values is $4.5$, the noise follows $Lap(0, 4.5/\epsilon)$. As before, we give a smaller weight to the noisy rating vectors and this is determined by cross validation. Second, [5] proposed a notion of "local privacy" in which differential privacy is guaranteed for each user separately. An optimal strategy for $d$-dimensional multinomial distribution in this case reduces effective samples from $n$ to $n\epsilon^2/d$ where $\epsilon$ is the DP parameter. In our case the dimension corresponds to the number of items

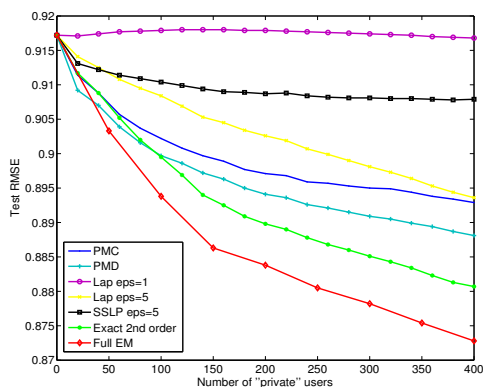

Figure 2: Test RMSE as a function of private user numbers. **PMC**: the privacy mechanism for continuous values; **PMD**: the privacy mechanism for discrete values; **Lap eps=1**: DP with Laplace noise, $\epsilon = 1$; **Lap eps=5**: same as before except $\epsilon = 5$; **SSLP eps=5**: sampling strategy described in [4] with DP parameter $\epsilon = 5$; **Exact 2nd order**: with exact second order statistics from private users (not a valid privacy mechanism); **Full EM**: EM without any privacy protection.

making estimation challenging under DP constraints. We also compare to this method and denote it as SSLP (sampling strategy for local privacy).

In addition, to understand how our approximation to second order statistics affects the performance, we also compare to the case that $r'a$ is given to the server directly where $a = \{-1, 1\}$ with probability $1/2$. In this way, the server can obtain the exact second order statistics using $r'r'^T$. Note that this is not a valid privacy preserving mechanism. Finally, we compare to the case that the algorithm can access private user rating vectors multiple times and update the parameters iteratively. Again, this is not a valid mechanism but illustrates how much could be gained.

Figure 2 shows the performance as a function of the number of private users. The standard Laplace noise method performs reasonably well when $\epsilon = 5$, however the corresponding privacy guarantee is very weak. SSLP improves the accuracy mildly.

In contrast, with the privacy mechanism we defined in section 4 the test RMSE decreases significantly as more private users are added. If we use the exact second order information without the sampling method, the final test RMSE would be reduced by $0.07$ compared to PMD. Lastly, full EM without privacy protection reduces the test RMSE by another $0.08$. These performance gaps are expected because there is an inherent trade-off between accuracy and privacy.

# 6 Conclusion

Our contributions in this paper are three-fold. First, we provide explicit guarantees for estimating item features in matrix completion problems. Second, we show how the resulting estimates, if shared with new users, can be used to predict their ratings depending on the degree of overlap between their private ratings and the relevant item subspace. The empirical results demonstrate that only a small number of public users with large number of ratings suffices for a good performance. Third, we introduce a new privacy mechanism for releasing 2nd order information needed for estimating item features while maintaining 1st order deniability. The experiments show that this mechanism indeed performs well in comparison to other mechanisms. We believe that allowing different levels of privacy is an exciting research topic. An extension of our work would be applying the privacy mechanism to the learning of graphical models in which 2nd or higher order information plays an important role.

# 7 Acknowledgement

The work was partially supported by Google Research Award and funding from Qualcomm Inc.

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
