[Supplementary Material]

## Supplementary Material

**Lemma 3.1.** *Assume $\epsilon_{i,j}$ for $(i,j) \in \Omega$ are i.i.d. sub-gaussian with $\sigma = \|\epsilon_{i,j}\|_{\psi_1}$. Then with probability $1 - \frac{e}{N^{4ch}}$, $\|\pi_B(\Delta)\|_* \leq \|\pi_{\overline{B}}(\Delta)\|_* + \frac{2c^2\sigma^2\sqrt{mn}}{N\lambda}\log^2 N$. Here $h > 0$ is an absolute constant associated with the sub-gaussian noise.*

*Proof.* Because $\hat{X}$ is the optimal solution to Eq.(2),

$$\frac{1}{N}\|Y - \hat{X}\|_{\Omega}^2 + \frac{\lambda}{\sqrt{mn}}\|\hat{X}\|_* \leq \frac{1}{N}\|Y - \mathring{X}\|_{\Omega}^2 + \frac{\lambda}{\sqrt{mn}}\|\mathring{X}\|_* \tag{10}$$

Plug in $Y_{i,j} = X_{i,j} + \epsilon_{i,j}$ and reorganize the terms, we get

$$\frac{1}{N}\|\Delta\|_{\Omega}^2 - \frac{1}{N}\sum_{(i,j)\in\Omega}\epsilon_{i,j}\Delta_{i,j} \leq \frac{\lambda}{\sqrt{mn}}(\|\mathring{X}\|_* - \|\mathring{X} + \Delta\|_*) \tag{11}$$

From our assumptions, the noise $\epsilon_{i,j} \sim Sub(\sigma^2)$. From a Hoeffding-type inequality (see proposition 5.10 in [15]), with probability at least $1 - \frac{e}{N^{4ch}}$,

$$\sum_{(i,j)\in\Omega}\epsilon_{i,j}\Delta_{i,j} \leq 2c\sigma\log N\|\Delta\|_{\Omega} \tag{12}$$

where $h > 0$ is a constant. Combining the two results yields

$$\frac{1}{N}\|\Delta\|_{\Omega}^2 \leq \frac{2\lambda}{\sqrt{mn}}(\|\mathring{X}\|_* - \|\mathring{X} + \Delta\|_*) + \frac{4c^2\sigma^2\log^2 N}{N} \tag{13}$$

By triangle inequality,

$$\|\mathring{X} + \Delta\|_* \geq \|\pi_A(\mathring{X}) + \pi_B(\Delta)\|_* - \|\pi_{\bar{A}}(\mathring{X})\|_* - \|\pi_{\bar{B}}(\Delta)\|_* \tag{14}$$

We assume $\mathring{X}$ has low rank and $A$, $B$ are two orthogonal subspace, then

$$\|\mathring{X} + \Delta\|_* \geq \|\mathring{X}\|_* + \|\pi_B(\Delta)\|_* - \|\pi_{\bar{B}}(\Delta)\|_* \tag{15}$$

Substitute the result back to (13) and combine with fact that $\|\Delta\|_{\Omega}^2 \geq 0$, we have

$$\|\pi_B(\Delta)\|_* \leq \|\pi_{\overline{B}}(\Delta)\|_* + \frac{2c^2\sigma^2\sqrt{mn}}{N\lambda}\log^2 N \tag{16}$$

$\square$

**Theorem 3.2.** *Assume RSC for the set $D(b, n, N)$ with parameter $\kappa > 0$ where $b = \frac{c\sigma\sqrt{m}}{\lambda_0\alpha}$. Let $\lambda = \lambda_0 c\sigma\frac{\log N}{\sqrt{N}}$, then we have $\frac{1}{\sqrt{mn}}\|\Delta\|_F \leq 2c\sigma(\frac{1}{\sqrt{\kappa}} + \frac{\sqrt{2r}}{\kappa})\frac{\log N}{\sqrt{N}}$ with probability at least $1 - \frac{e}{N^{4ch}}$ where $h, c > 0$ are constants.*

*Proof.* Combining (13) and (12) yields,

$$\frac{1}{N}\|\Delta\|_{\Omega}^2 \leq \frac{2\lambda}{\sqrt{mn}}\|\pi_{\bar{B}}(\Delta)\|_* + \frac{4c^2\sigma^2\log^2 N}{N} \leq \frac{2\sqrt{2r}\lambda}{\sqrt{mn}}\|\Delta\|_F + \frac{4c^2\sigma^2\log^2 N}{N} \tag{17}$$

where the second inequality is because the rank of $\pi_{\bar{B}}(\Delta)$ is at most $2r$. From RSC property,

$$\frac{k}{mn}\|\Delta\|_F^2 \leq \frac{2\sqrt{2r}\lambda}{\sqrt{mn}}\|\Delta\|_F + \frac{4c^2\sigma^2\log^2 N}{N} \tag{18}$$

The bound on $\Delta\|_F$ can be subsequently obtained by solving a quadratic equation, and we get

$$\frac{1}{\sqrt{mn}}\|\Delta\|_F \leq \frac{2c\sigma\log N}{\sqrt{\kappa N}} + 2\sqrt{2r}\frac{\lambda}{\kappa} \tag{19}$$

Substituting $\lambda$ completes the proof. $\square$

**Lemma 3.3.** *If $\frac{1}{\sqrt{mn}}\|\hat{X} - \mathring{X}\|_F \leq \delta$, then $\|\hat{\Sigma} - \mathring{\Sigma}\|_F \leq \sqrt{2}\delta$*

*Proof.* Let $PSQ^T$ be the SVD of $\sqrt{R}X$, then upon a unitary transformation $U = R^{-\frac{1}{2}}PS^{\frac{1}{2}}$ and $V = QS^{\frac{1}{2}}$. Correspondingly $\Sigma = \frac{1}{\sqrt{mn}}VV^T = \frac{1}{\sqrt{mn}}QSQ^T$. Note that $QSQ^T$ is equivalent to the polar decomposition of $X$ defined as $(X^TX)^{\frac{1}{2}}$. The perturbation bound for polar decomposition (see [9]) then gives

$$
\begin{aligned}
\|\Sigma - \mathring{\Sigma}\|_F &\leq \sqrt{\frac{2}{mn}}\|\sqrt{R}X - \sqrt{R}\mathring{X}\|_F \\
&= \sqrt{2}\|X - \mathring{X}\|_F
\end{aligned}
\tag{20}
$$

$\square$

**Theorem 3.4.** *Assume $\mathring{X}_i$ are i.i.d samples from a distribution with support only in a subspace of dimension $r$ and bounded norm $\|\mathring{X}_i\| \leq \alpha\sqrt{m}$. Let $\beta_1$ and $\beta_r$ be the smallest and largest eigenvalues of $\Sigma^*$. Then, for large enough $n$, with probability at least $1 - \frac{r}{n^2}$,*

$$
\|\mathring{\Sigma} - \Sigma^*\|_F \leq 2\sqrt{r}\alpha\sqrt{\frac{\beta_r \log n}{\beta_1 n}} + o(\frac{\log n}{n})
\tag{21}
$$

*Proof. (sketch)* We first bound $\|\Theta^n - \Theta^*\|_{op} \leq 2\alpha\sqrt{\frac{\beta_r \log n}{n}}$ by matrix Bernstein inequalities. If $\beta_1$ is the minimum eigenvalue, $\|\mathring{\Sigma} - \Sigma^*\|_{op} = \|(\Theta^n)^{\frac{1}{2}} - (\Theta^*)^{\frac{1}{2}}\|_{op}$ is upper bounded by $\frac{1}{2\beta_1}\|\Theta^n - \Theta^*\|_{op}$. Finally we have used $\|\mathring{\Sigma} - \Sigma^*\|_F \leq \sqrt{r}\|\mathring{\Sigma} - \Sigma^*\|_{op}$. $\square$

Combining this with the previous theorem, we find that for large enough $n$ and $c$, with probability at least $1 - \frac{2r}{n^2}$,

$$
\begin{aligned}
\|\hat{\Sigma} - \Sigma^*\|_F &\leq c\sigma\sqrt{\frac{\log N}{N}}(\sqrt{\frac{6}{\kappa}} + \frac{4\sqrt{2r}\lambda_0\alpha}{\kappa}) \\
&+ 2\sqrt{r}\alpha\sqrt{\frac{\beta_r \log n}{\beta_1 n}} + o(\frac{\log n}{n})
\end{aligned}
\tag{22}
$$

**Theorem 3.5.** *Suppose $\|\Sigma - \Sigma^*\|_F \leq \delta \ll \beta_1$, $\phi \in \mathcal{D}(\gamma)$. For any $\mathring{x} \in row((Q^*)^T)$, our observation $x_\phi = \mathring{x}_\phi + \epsilon_\phi$ where $\epsilon_\phi \sim Sub(\sigma^2)$ is the noise vector. The predicted ratings over the remaining entries are given by $\hat{x}_{\phi^c} = \Sigma_{\phi^c,\phi}(\lambda'I + \Sigma_{\phi,\phi})^{-1}x_\phi$. Then, with probability at least $1 - \exp(-c_2 \min(c_1^4, \sqrt{|\phi|}c_1^2))$,*

$$
\|x_{\phi^c} - \mathring{x}_{\phi^c}\|_F \leq 2\sqrt{\lambda' + \delta}(\sqrt{\gamma\frac{m}{|\phi|}} + 1)(\frac{\|\mathring{x}\|_F}{\sqrt{\beta_1}} + \frac{2c_1\sigma|\phi|^{\frac{1}{4}}}{\sqrt{\lambda'}})
$$

*where $c_1, c_2 > 0$ are constants.*

*Proof. (sketch)* We consider an equivalent optimization problem

$$
\hat{x}_{\phi^c} = \text{argmin}_y \ [x_\phi; y]^T(\lambda'I + \Sigma)^{-1}[x_\phi; y]
\tag{23}
$$

where the coordinates $\phi$ are arranged on top. From optimality of $\hat{x}_{\phi^c}$ in comparison to $y = \mathring{x}_{\phi^c}$ we can upper bound $\|[0; \hat{x}_{\phi^c} - \mathring{x}_{\phi^c}]^T(\lambda'I + \Sigma)^{-\frac{1}{2}}\|_F$ (*) in terms of $\|\mathring{x}\|$ and a noise term. The difference vector $[0; \hat{x}_{\phi^c} - \mathring{x}_{\phi^c}]$ can be decomposed as $[x_1; y_1] + [-x_1; y_2]$ where $[x_1, y_1]^T \in col(Q^*)$ and $[-x_1; y_2]^T \in col((Q^*)^{\perp})$ and the goal is to upper bound them separately. By triangle inequality, we have $[0; \hat{x}_{\phi^c} - \mathring{x}_{\phi^c}]^T(\lambda'I + \Sigma)^{-\frac{1}{2}}\|_F \geq \|[-x_1; y_2]^T(\lambda'I + \Sigma)^{-\frac{1}{2}}\|_F - \|[x_1; y_1]^T(\lambda'I + \Sigma)^{-\frac{1}{2}}\|_F$. For a small $\delta$ we can get upper and lower bounds on $(\lambda I + \Sigma)^{-1}$ projected to the two orthogonal subspaces. Applying these bounds to the previous inequality and (*), we can bound $\|[-x_1; y_2]\|_F$ as well as $\|x_1\|_F$. Finally, since $\phi \in D(\gamma)$ we can estimate $\|[x_1; y_1]\|_F$ as a function of $\|x_1\|_F$. The result follows by combining these and triangle inequality $\|x_{\phi^c} - \mathring{x}_{\phi^c}\|_F \leq \|[x_1; y_1]\|_F + \|[-x_1; y_2]\|_F$. $\square$

**Theorem 4.2.** *Let $z_i = sign(r'_i)$ and $h = (\sum_{i=1}^{m} |r'_i|)/m$. The minimizing distribution of (9) is given by $Pr(u = zh) = Pr(u = -zh) = 1/2$.*

*Proof.* Expanding the objective yields

$$E_P \| uu^T - r'r'^T \|_F^2$$

$$= E_P \sum_{i,j} \left\{ u_i^2 u_j^2 - 2u_i u_j r'_i r'_j + r'^2_i r'^2_j \right\}$$

$$= E[(\sum_i u_i^2)^2] - 2E[(\sum_i u_i r'_i)^2] + (\sum_i r'^2_i)^2$$

From the privacy constraint, $E[(\sum_i u_i^2)^2] \geq E^2[\sum_i u_i^2] = m^2 v^2$. In addition, because $E[u_i u_j] r'_i r'_j \leq \sqrt{E[u_i^2]E[u_j^2]}|r'_i r'_j| = v|r'_i r'_j|$, $E[(\sum_i u_i r'_i)^2] \leq v(\sum_i |r'_i|)^2$. Therefore the objective is lower bounded as

$$m^2 v^2 - 2v(\sum_i |r'_i|)^2 + (\sum_i r'^2_i)^2$$

The lower bound is attained when $u = \sqrt{v}z\, a$, where $a$ is a single binary random variable taken values in $\{-1, +1\}$ with equal probability. Finally, optimizing $v$ gives $v = (\frac{\sum_i |r'_i|}{m})^2$.

$\square$