[Reviews · NeurIPS 2014]

Submitted by Assigned_Reviewer_16

This paper proposes a privacy-preserving mechanism for recommender systems
which are based on the popular matrix factorization. Specifically, the
authors propose splitting users into disjoint groups of public and private
users. Private users do not share any information (ratings) with the
system while public users share all of their information. The authors show
that under certain technical conditions one can bound the estimation
accuracy of the item features based on the number of observed ratings.
This estimate can further be used to bound the reconstruction error of the
private users. The authors further demonstrate the empirical performance
of their approach using the Movielens 10M dataset.

Privacy in recommender systems is an important problem that we are only
beginning to explore. As far as I know most formal previous approaches
rely on differential privacy. This paper considers that even the
recommender engine is not safe. This seems like a reasonable practical
setting. In that setting the authors develop an interesting framework.
The formalism developed in this paper is clear and seems sound as are the
derivations. The results (bounding the error of the item factors and
consequently the errors of the reconstruction) are interesting. It would
be nice to give a bit more intuition about Theorem 3.5.

Overall the paper is well written and easy to follow.

The experiments are also reasonable. They demonstrate what is needed and
the comparisons to other methods is reasonably convincing.

In the first set of experiments, it would be good to clarify what the
label "Percentage of Users" mean. I understood it to mean the percentage
of all users that were public.

The second set of experiments uses 100 public users and up to 400 private
users. It is a bit unclear why the authors are not reporting results with
more private uses (the dataset contains 10K users). What happens in that
setting? Do any of the DP methods (especially LAP eps=5) reach similar
performance as PMC and PMD. Was there a reason for stopping at 400 private
users? It would be good to show that PMC and PMD do well in a less
synthetic setting. It would also be nice to provide results in the case
where the private users aren't the ones that have necessarily consumed the
most items.
Summary: Good paper which introduces an interesting way to preserve privacy in
matrix-factorization-based recommender systems. Both theoretical analysis
and empirical results seem sound.

Submitted by Assigned_Reviewer_35

This paper addresses the high sparsity problem of user-item ratings in recommendation systems, when applying matrix factorization technique. Some users own a large number of ratings (called public users in the paper), whereas many users have very few ratings (called private users). A main reason for the latter case is that such private users intend to protect their privacy.

First, The authors provided theoretical guarantees of reasonably good performance of recommendation systems, when only a small set of public users are available. Then, they defined mechanisms for private protection for both discrete and continuous rating values to exploit private users in order to improve the models initially built on public users' ratings. The mechanisms add private users by maintaining the second order statistics of the user ratings.
Finally, they tested the proposed private preserving mechanisms on the MovieLens 10M data set. The experimental results confirmed the promised guarantees and showed improvements by exploring private users.

This paper is clearly written. The novelty of the paper lies in the theoretical proof of the performance guarantee and the definition of the private protection mechanisms. Experiments are somewhat limited
Summary: A paper with novel ideas, supported by proofs and experimental results

Submitted by Assigned_Reviewer_43

This paper proposes an EM-like algorithm for matrix completion applied to recommendation systems; parameter estimation is constrained in order to maintain privacy guarantees. The authors modify the typical matrix completion with trace-norm regularization to only estimate the item features.

Quality:
The paper illustrates a nice use of privacy aware learning for the application of recommendation systems. They show that a small number of public users having a large number of ratings can provide sufficient overlap with private data to enable good accuracy. They use the private data to estimate covariances, while keeping a particular marginal distribution that helps maintain privacy.

Clarity:
The paper is well written. A few things could have been elaboarated:
- It's not quite clear how their method compares to previous methods, e.g. [10], either experimentally or in terms of privacy guarantees.
- It would be nice to show summary statistics/plots of the marginal distributions to help illustrate the affects of their technique.

Significance:
This paper belongs to an important class of algorithms that allow one to choose between privacy and accuracy. If data privacy continues to be in the public spotlight, this paper could be a nice addition to that field.

Originality:
To this reader's knowledge, their approach is novel, borrowing from common techniques in privacy aware learning.
Summary: The paper illustrates a nice application of privacy aware learning to recommendation systems. Further experiments would strengthen the reader's understanding of how the algorithm performs, whether it meets its privacy goals, and how it compares to previous methods.
Author Feedback
Author rebuttal: We thank all the reviewers for their comments.

Reviewer 16:

Yes, “percentage of users” refers to the percentage of users that are public.

Our second experiment shows that PMC and PMD outperform DP in the presence of a small number of private users. We didn’t use all the users in this experiment because the EM algorithm for DP and others requires us to fill in (obfuscate) the full rating vector for each user. This becomes expensive on the server side. The issue can be dealt with on-line algorithms or by using multiple servers.

Using private users with most ratings shows the performance differences more robustly. The results remain close if we select private users uniformly.

Reviewer 35:

We agree additional experiments would only strengthen the paper. However, the two experiments included highlight the main results: 1) the error bound and 2) the new privacy mechanism. Consistent with our bound, we show that a small number of public users suffice for good accuracy. In the second part, we compare our privacy mechanism to “local privacy” proposed in [5]. Our algorithm (privacy concept) significantly outperforms theirs.

Reviewer 43:

Our setting differs substantially from [10] where they assume that a trusted server holds all the data. In [10], the goal is to publish statistics so as to make recommendations to individual users. This is a standard DP setting. In contrast, our server is untrusted and each user must guard their own data. DP does not apply directly. “local privacy” in [5] extends DP to our setting, and we provide a comparison in figure 2.

We will add a summary plot of marginal distributions to the supplementary material.